# Comparative Evaluation of Plasma Metabolomic Data from Multiple Laboratories

**DOI:** 10.3390/metabo12020135

**Published:** 2022-02-01

**Authors:** Shin Nishiumi, Yoshihiro Izumi, Akiyoshi Hirayama, Masatomo Takahashi, Motonao Nakao, Kosuke Hata, Daisuke Saigusa, Eiji Hishinuma, Naomi Matsukawa, Suzumi M. Tokuoka, Yoshihiro Kita, Fumie Hamano, Nobuyuki Okahashi, Kazutaka Ikeda, Hiroki Nakanishi, Kosuke Saito, Masami Yokota Hirai, Masaru Yoshida, Yoshiya Oda, Fumio Matsuda, Takeshi Bamba

**Affiliations:** 1Department of Omics Medicine, Hyogo College of Medicine, 1-1 Mukogawa-cho, Nishinomiya-city, Hyogo 663-8501, Japan; 2Division of Metabolomics, Medical Institute of Bioregulation, Kyushu University, 3-1-1 Maidashi, Higashi-ku, Fukuoka-city, Fukuoka 812-8582, Japan; m-takahashi@bioreg.kyushu-u.ac.jp (M.T.); nakao@bioreg.kyushu-u.ac.jp (M.N.); k-hata@bioreg.kyushu-u.ac.jp (K.H.); bamba@bioreg.kyushu-u.ac.jp (T.B.); 3Institute for Advanced Biosciences, Keio University, 246-2 Mizukami, Kakuganji, Tsuruoka-city, Yamagata 997-0052, Japan; 4Laboratory of Biomedical and Analytical Sciences, Faculty of Pharma-Science, Teikyo University, 2-11-1 Kaga, Itabashi-ku, Tokyo 173-8605, Japan; saigusa.daisuke.vf@teikyo-u.ac.jp; 5Department of Integrative Genomics, Tohoku Medical Megabank Organization, Tohoku University, 2-1 Seiryo-machi, Aoba-ku, Sendai-city, Miyagi 980-8573, Japan; ehishi@ingem.oas.tohoku.ac.jp (E.H.); matsukawa@megabank.tohoku.ac.jp (N.M.); 6Advanced Research Center for Innovations in Next-Generation Medicine, Tohoku University Tohoku Medical Megabank Organization, 2-1 Seiryo-machi, Aoba-ku, Sendai-city, Miyagi 980-8573, Japan; 7Department of Lipidomics, Graduate School of Medicine, The University of Tokyo, 7-3-1 Hongo, Bunkyo-ku, Tokyo 113-0033, Japan; stokuoka@m.u-tokyo.ac.jp (S.M.T.); kita@m.u-tokyo.ac.jp (Y.K.); fhamano@lscore.m.u-tokyo.ac.jp (F.H.); yoda@m.u-tokyo.ac.jp (Y.O.); 8Life Sciences Core Facility, Graduate School of Medicine, The University of Tokyo, 7-3-1 Hongo, Bunkyo-ku, Tokyo 113-0033, Japan; 9Department of Bioinformatic Engineering, Graduate School of Information Science and Technology, Osaka University, 1-5 Yamadaoka, Suita-city, Osaka 565-0871, Japan; n-okahashi@ist.osaka-u.ac.jp (N.O.); fmatsuda@ist.osaka-u.ac.jp (F.M.); 10Laboratory of Biomolecule Analysis, Department of Applied Genomics, Kazusa DNA Research Institute, 2-6-7 Kazusa-Kamatari, Kisarazu-city, Chiba 292-0818, Japan; kaikeda@kazusa.or.jp; 11Lipidome Lab Co., Ltd., 1-2 Nukazuka, Yanagida, Akita-city, Akita 010-0825, Japan; hnakani@lipidome.jp; 12Division of Medical Safety Science, National Institute of Health Sciences, 3-25-26 Tonomachi Kawasaki-ku, Kawasaki-city, Kanagawa 210-9501, Japan; saitok2@nihs.go.jp; 13RIKEN Center for Sustainable Resource Science, 1-7-22 Suehiro-cho, Tsurumi-ku, Yokohama-city, Kanagawa 230-0045, Japan; masami.hirai@riken.jp; 14Department of Food Science & Nutrition, Research Institute of Food and Nutritional Sciences, Graduate School of Human Science & Environment, University of Hyogo, 1-1-12 Shinzaikehon-cho, Himeji-city, Hyogo 670-0092, Japan; myoshida@shse.u-hyogo.ac.jp

**Keywords:** metabolomics, relative quantification, inter-laboratory comparison, mass spectrometry, hydrophilic metabolite, hydrophobic metabolite

## Abstract

In mass spectrometry-based metabolomics, the differences in the analytical results from different laboratories/machines are an issue to be considered because various types of machines are used in each laboratory. Moreover, the analytical methods are unique to each laboratory. It is important to understand the reality of inter-laboratory differences in metabolomics. Therefore, we have evaluated whether the differences in analytical methods, with the exception sample pretreatment and including metabolite extraction, are involved in the inter-laboratory differences or not. In this study, nine facilities are evaluated for inter-laboratory comparisons of metabolomic analysis. Identical dried samples prepared from human and mouse plasma are distributed to each laboratory, and the metabolites are measured without the pretreatment that is unique to each laboratory. In these measurements, hydrophilic and hydrophobic metabolites are analyzed using 11 and 7 analytical methods, respectively. The metabolomic data acquired at each laboratory are integrated, and the differences in the metabolomic data from the laboratories are evaluated. No substantial difference in the relative quantitative data (human/mouse) for a little less than 50% of the detected metabolites is observed, and the hydrophilic metabolites have fewer differences between the laboratories compared with hydrophobic metabolites. From evaluating selected quantitatively guaranteed metabolites, the proportion of metabolites without the inter-laboratory differences is observed to be slightly high. It is difficult to resolve the inter-laboratory differences in metabolomics because all laboratories cannot prepare the same analytical environments. However, the results from this study indicate that the inter-laboratory differences in metabolomic data are due to measurement and data analysis rather than sample preparation, which will facilitate the understanding of the problems in metabolomics studies involving multiple laboratories.

## 1. Introduction

Mass spectrometry (MS)-based metabolomics have been widely applied to a variety of research fields, targeting microorganisms, plants, animals, and humans, to understand metabolism in the body [1,2,3,4]. Additionally, the relatively large datasets obtained from metabolomics are treated in large-scale studies, such as cohort and epidemiological studies [5,6,7]. When small-scale studies based on MS-based metabolomics are carried out, it is sufficient to proceed with metabolomics in one laboratory; in these cases, there is no particular problem with using only one mass spectrometer or protocols optimized at one laboratory. However, large-scale studies often require metabolomic analyses using various types of mass spectrometers in multiple laboratories. In these situations, data integration between laboratories is an issue to be considered. It has been reported that the use of different metabolomics platforms in biomarker studies based on non-targeted metabolomics results in low confidence in the reproducibility of the results [8].

Investigations of the differences in MS-based metabolomics from different laboratories/machines have been reported by some research groups. For example, Siskos et al. have investigated the inter-laboratory differences in targeted metabolomics using AbsoluteIDQ, which is employed to carry out quantitative analyses of amino acids, amines, acylcarnitines, glycerophospholipids, and sphingolipids. More than 80% of plasma metabolites have a relative standard deviation (RSD%) of <20% between laboratories using the same protocols [9]. The quantitative lipidomics analysis by Bowden et al. has shown that the plasma concentration of many lipids differs between the laboratories using non-normalized protocols [10]. Both studies performed quantitative analysis, but not semi-quantitative analysis, which has been applied to metabolomics studies in many research groups. The main difference is the standardization of the analytical protocols, and Siskos et al. used the same analysis kit at all laboratories, although the chromatographs and mass spectrometers used in each laboratory are different. Thus, the standardization of the analytical protocols is important to resolve the inter-laboratory differences. Further, Liebisch et al. advocated the necessity of the “gold-standard” protocols for metabolomics [11]. Several other investigations of inter-laboratory differences in metabolomics are reported [12,13,14]. However, it is very difficult to create the gold standard protocols in metabolomics because the types of chromatographs and mass spectrometers that can be used in each laboratory are different.

In our previous study by Izumi et al. [15], the extracted samples obtained from two types of cultured cells are transferred to some laboratories, and metabolomics is performed using the daily routine protocols of each laboratory, including the additional metabolite extraction steps. This means that each laboratory performed metabolomics without the standardization of the analytical protocols. The results showed that only approximately 60% of the metabolites detected in the laboratories were not different. In this study, metabolomics for identically extracted samples prepared from human and mouse plasma is carried out at some laboratories; however, unlike our previous study, the additional metabolite extraction steps are not conducted [15]. Additionally, a strict quantitative confirmation step is added in this evaluation, and the current article is a sequential study that is developed from our previous study [15], and the goal of this study is to understand the reality of inter-laboratory differences in metabolomics, including the differences in analytical methods, as potential the problems in metabolomics studies involving multiple laboratories. For example, from this study, it may be possible to find out which metabolites lead to different results depending on the analytical methods.

## 2. Results and Discussion

### 2.1. Analytical Procedure and Data Acquisition

The analytical procedure of this study is shown in Figure 1; nine laboratories participated in this study. In this study, identically extracted samples were prepared from two kinds of plasma (human and mouse plasma), and their mixed plasma was distributed to nine participating laboratories throughout Japan. The relative quantification of each metabolite in the samples was performed using different targeted metabolomics methods using several separation techniques (gas chromatography, GC; liquid chromatography, LC; capillary electrophoresis, CE; ion chromatography, IC; and supercritical fluid chromatography, SFC) coupled with MS (quadrupole mass spectrometry, QMS; time-of-flight mass spectrometry, TOFMS; quadrupole-time-of-flight mass spectrometry, QTOFS; triple quadrupole mass spectrometry, TQMS; quadrupole-Orbitrap mass spectrometry, Q Exactive; and quadrupole-Orbitrap-linear ion trap mass spectrometry, Orbitrap Fusion). In all the laboratories, the peak for each metabolite was identified according to their routine analytical protocols, and each peak area/height value was measured. No protocol was shared; however, a list of targeted metabolites was prepared based on our previous report [15]. For detailed information on the metabolites targeted in this study, we consulted our previous report [15]. In each laboratory, metabolomics was carried out without conducting any work other than the pretreatment required for each analysis, and 18 analytical methods were employed to extract each sample in triplicate.

The identification of hydrophilic metabolites was carried out according to the protocol of each laboratory, including the comparison of the retention/migration time, MS and tandem MS (MS/MS) spectra, and the multiple reaction monitoring (MRM) transition of the targeted metabolites in the samples with those of authentic standards analyzed under identical conditions. For the hydrophobic metabolites (lipids) analysis, we targeted 21 lipid classes, including acylcarnitine (AC), cholesterol ester (ChE), ceramide (NS) (Cer(NS)), diacylglycerol (DG), free fatty acid (FA), hexose ceramide (NS) (HexCer(NS)), lysophosphatidic acid (LPA), lysophosphatidylcholine (LPC), lysophosphatidylethanolamine (LPE), lysophosphatidylglycerol (LPG), lysophosphatidylinositol (LPI), lysophosphatidylserine (LPS), monoacylglycerol (MG), phosphatidic acid (PA), phosphatidylcholine (PC), phosphatidylethanolamine (PE), phosphatidylglycerol (PG), phosphatidylinositol (PI), phosphatidylserine (PS), sphingomyelin (SM), and triacylglycerol (TG), with a variety of 20 FA side chains. Thus, the targeted lipid molecules comprised 3390 compounds. The identification of lipids was conducted based on the retention time, precursor ion, and fragmentation patterns or specific MRM transitions of each metabolite.

The information on the analytical methods for detecting the hydrophilic and hydrophobic metabolites are shown in Table 1 and Table 2. Eleven methods for the hydrophilic metabolites from the six laboratories and seven methods for the hydrophobic metabolites from the six laboratories were used. The detailed analytical information is shown in Appendix A, so that the differences among the laboratories can be understood. To compare the results from the laboratories, the relative signal intensity data of the metabolites were subjected to statistical evaluation. The evaluations were performed in two steps (first and second steps). The first step proceeded with the same procedures as in our previous report [15] for all metabolites detected. In the second step, the metabolites guaranteed to be quantitatively analyzed were selected from all the metabolites detected based on the values of [S-2]/([S-1] + [S-3]) × 100, as described in Section 3.2 and Section 3.4, the sample preparation and data analysis sections. Afterward, the same evaluations in the first step were performed on the selected metabolites.

### 2.2. Data Summary and Comparisons of the Analytical Methods for the Relative Quantification for the Metabolites Detected (First Step)

First, 160 hydrophilic and 660 hydrophobic metabolites were identified in human and mouse plasma using at least one analytical method (Figure 2, Table 3). Among these metabolites, 111 hydrophilic and 291 hydrophobic metabolites in both human and mouse plasma were detected by ‘two or more’ analytical methods (Table 3). Using ‘three or more’ analytical methods, 74 hydrophilic and 188 hydrophobic metabolites were detected (Appendix A). Using ‘four or more’ analytical methods, 51 hydrophilic and 114 hydrophobic metabolites were detected (Appendix A). Further, 402 metabolites were identified by multiple analytical methods, and the proportion of all the metabolites identified in at least one analytical method was almost the same as that in our previous study [15].

The relative ratios (human/mouse) of 160 hydrophilic metabolites obtained using the 11 different methods and 660 hydrophobic metabolites obtained using the 7 different methods are shown in Appendix A, respectively. In the metabolites identified by ‘two or more’ analytical methods, the statistical significance of each metabolite between human and mouse plasma samples was determined using a two-sided Student’s *t*-test (*p* < 0.05). From the results, 88 hydrophilic and 256 hydrophobic metabolites were statistically significant among the human and mouse plasma samples (Table 3). Among the significantly different metabolites, the numbers (percentages) of hydrophilic and hydrophobic metabolites with similar human/mouse values from multiple methods were 82 (93.2%) and 243 (94.9%), respectively (Table 3). This result means that the percentages of metabolites with different results among the laboratories were only 6.8% for the hydrophilic metabolites and 5.1% for the hydrophobic metabolites. This suggested that most metabolites could be qualitatively analyzed in each laboratory.

The one-way ANOVA using human/mouse values (α = 0.05) showed that there was no difference between the 40 hydrophilic and 62 hydrophobic metabolites across multiple methods (Table 3). Further, 102 metabolites corresponded to 25.4% (102/402) of the 402 metabolites identified from both samples by ‘two or more’ analytical methods (Table 3). These results indicate that the proportion of metabolites with similar human/mouse values from the laboratories was approximately one-quarter. However, when ignoring one outlier in the data, the results improved: the human/mouse (S-1/S-3) levels were similar among the 56 hydrophilic and 135 hydrophobic metabolites, and the total 191 metabolites corresponded to 47.5% (191/402) of the 402 metabolites identified from both samples by ‘two or more’ analytical methods (Table 3). These evaluation procedures based on ‘two or more’ analytical methods are the same as those performed in our previous study [15], and the differences between the current and previous studies are the presence or absence of pre-treatment, including the metabolite extraction before measurement, and the sample types. In addition, the number of analytical methods was also different. An important point in the current study is that there are few differences in the pretreatment procedures among the laboratories. These results indicate that the differences in separation methods, such as GC, LC, CE, IC, and SFC, may contribute to inter-laboratory differences in the results of MS-based metabolomics. Additionally, in all the evaluations using the metabolites detected by ‘two or more’, ‘three and more’, and ‘four or more’ analytical methods, the proportion of metabolites without the inter-laboratory differences was higher for hydrophilic metabolites than for hydrophobic metabolites (Appendix A). In contrast, the differences between the hydrophilic and hydrophobic metabolites were not as large in our previous study [15]. Therefore, the differences in the types of machines, such as GC/MS, LC/MS, CE–MS, IC/MS, and SFC/MS, can affect the inter-laboratory differences in the measurement results of hydrophobic metabolites.

### 2.3. Data Summary and Comparisons of the Analytical Methods for the Relative Quantification of the Quantitatively Guaranteed Metabolites (Second Step)

MS-based metabolomics facilitates the simultaneous measurement of numerous metabolites; however, some metabolites may be measured with low quantitativeness in certain analytical methods because of the dynamic ranges and detection sensitivity of mass spectrometers. Therefore, in the second step, the metabolites that were quantitatively measured were selected from all the detected metabolites; thereafter, we performed the first step on the quantitatively guaranteed metabolites (Figure 1). In this study, the quantitatively guaranteed metabolites were selected based on the values of [S-2]/([S-1] + [S-3]) × 100, as described in Section 3.4. Consequently, 131 hydrophilic and 297 hydrophobic metabolites were selected based on the criteria set above. The percentage of metabolites selected was 81.9% (131/160) for the hydrophilic metabolites and 45.0% (297/660) for the hydrophobic metabolites (Table 4). These results indicate that the quantitatively guaranteed percentage of hydrophilic metabolites may be higher than that of hydrophobic metabolites. In the evaluations by ‘two or more’ analytical methods, the one-way ANOVA using human/mouse values (α = 0.05) showed that there was no difference between the 30 hydrophilic and 49 hydrophobic metabolites across multiple methods (Table 4). Further, 79 metabolites corresponded to 32.9% (79/240) of the 240 metabolites identified from both samples by ‘two or more’ analytical methods (Table 4). These results indicate that the proportion of metabolites with similar human/mouse values among the laboratories was approximately one-third. However, when ignoring one outlier in the data, the results improved such that the human/mouse levels were similar among the 48 hydrophilic and 87 hydrophobic metabolites. Further, 135 metabolites corresponded to 56.3% (135/240) of the 240 metabolites identified from both samples by ‘two or more’ analytical methods (Table 4). The percentages of the metabolites without the inter-laboratory differences in the second step were relatively higher than those in the first step, which targets all the detected metabolites. Further, the improvement in these percentages was due to the results of hydrophobic metabolites (Table 3 and Table 4). A similar tendency was observed in the evaluations by ‘three or more’ and ‘four or more’ analytical methods (Appendix A).

### 2.4. Possible Causes of the Differences in the Results from Different Laboratories/Machines in MS-Based Metabolomics

#### 2.4.1. Hydrophilic Metabolites

Regarding the hydrophilic metabolites, the number of metabolites with inter-laboratory differences was relatively small compared to the hydrophobic metabolites (Table 3 and Table 4, Appendix A). In Figure 3A, the results of essential and non-essential amino acids, which are abundant in biological samples, including blood plasma, are shown. The patterns of the inter-laboratory differences in these amino acids under the analytical conditions of this study were different from those of our previous study [15]. Characteristic to the current study, the results from LC/MS with a pentafluorophenylpropyl column (Method E) were different from those from other methods, although this was not observed in our previous study [15]. This may mean that there are plasma-specific factors that affect plasma metabolomics in an analytical environment similar to Method E, such as the types of chromatographs/mass spectrometers and their analytical conditions. In contrast, large variability in the results of histidine (His) from the laboratories were observed between this study (Figure 3A) and our previous study [15]. Figure 3B shows the results of these amino acids when quantitatively guaranteed metabolites were selected in the second step. In this step, the inter-laboratory differences seemed to have disappeared to some extent, although some amino acids were excluded in the second step. This means that the analytical stability at each laboratory is also important.

Differences in the relative ratios (human/mouse) from multiple methods, including CE–MS (Methods A–B), IC/MS (Methods C–D), LC/MS (Methods E–F), and GC/MS (Methods G–K), were observed (Figure 4). The hydrophilic metabolites, including citric acid (Cit), creatinine, glucaric acid, glucuronate, and uric acid, are indicated in Figure 4, with remarkable differences from the multiple methods. In general, GC/MS requires derivatization pretreatments and is standardized using the electron ionization (EI) method. However, the mass spectrometers accompanied by chromatographs, except GC/MS, are not normalized to ionization conditions. Trimethylsilyl derivatization is not suitable for the measurement of amino acids by GC/MS because of its low derivatization efficiency; however, this was not observed in this study (Figure 3). No molecular subclass tendency was confirmed in the metabolites with differences in the separation and mass spectrometry machines (Figure 3), and these metabolites were not essential and non-essential amino acids. In addition to the derivatization pretreatments and EI-based methods, heat stability may be a key factor. Further, only human or mouse plasma may contain factors that affect the measurement by each chromatograph or mass spectrometer. For example, Cit (larger SD values and higher S-1/S-3 ratios in GC/MS compared with other analyses), creatinine (lower S-1/S-3 ratios in GC/MS compared with other analyses), and uric acid (larger SD values and higher S-1/S-3 ratios in GC/MS compared with other analyses) showed large differences in the separation and mass spectrometry machines. Furthermore, the presence of influential factors in human or mouse plasma was suspected. However, the reasons for these differences were difficult to determine from the results of this study.

#### 2.4.2. Hydrophobic Metabolites

In the hydrophobic metabolites, the differences in the results from LC/MS (Methods A–D) and SFC/MS (Methods E–F) were remarkable. These results are presented in Figure 5. Many of the relative ratios (human/mouse) were lower in SFC/MS (Methods E–F) than in LC/MS (Methods A–D). This means that chromatographs may affect the differences in the relative quantitation data between the laboratories, and similar suggestions were shown in the study by Chocholoušková et al. [29]. For example, the reverse-phase LC used in Methods A–D separated each lipid molecule based on the hydrophobic interactions between the nonpolar side chains of C8 or C18 particles and the hydrophobic fatty acyl chains of lipids. However, SFC separated each lipid class with the normal phase column used in Method E because the stationary phase with high polarity recognized the head group of lipids rather than fatty acyl chains. In general, a modern analytical SFC/MS system achieves discrimination among isobaric and isomeric compounds (e.g., surfactants and pesticides) through a combination of mass resolution and highly efficient separation by SFC [30,31]. In this study, in the SFC/MS method (Method E), all lipid molecules in the same class were eluted at similar retention times by SFC-based lipid class separation [28]. Because this technique (Method E) co-elutes the same class of lipids, the ion-suppression and/or ion-enhancement effects of the biological matrix can be normalized by adding the appropriate internal standards of each lipid class (stable isotope-labeled lipid standards) [19,28]. However, in this study, to clarify the cause of the inconsistency in the relative quantification (S-1/S-3) results from the analytical methods, the relative quantification values were calculated using the raw data (peak area or height) without correcting the acquired data with the internal standards. Additionally, the types and amounts of lipids in human and mouse plasma vary widely. Therefore, it has been suggested that the SFC/MS methods have different matrix effects (ionization suppression/enhancement) on the individual lipid molecules contained in each lipid class, depending on the sample species. Thus, the relative quantification results of the SFC/MS and LC/MS methods were different (Figure 5). To compare the quantitative accuracy of the SFC/MS and LC/MS data, it is necessary to calculate and compare the absolute amount of each lipid molecule using the stable isotope-labeled lipid standards of each lipid class [20]. Further, the inter-laboratory differences in plasma lipid analysis were reported by Bowden et al. [10]. In the study by Bowden et al., lipid analysis was accompanied by the daily routine protocols of each laboratory, which means that the protocols were not normalized. They were performed by 31 laboratories, and the differences in the levels of various plasma lipids among these laboratories were observed. The need to define the generally accepted guidelines for quantitative MS-based plasma/serum lipidomics, which allows the integration of data obtained from different instrumentation platforms across independent laboratories, has been proposed [32].

#### 2.4.3. Hydrophilic Metabolites versus Hydrophobic Metabolites

From the evaluations of this study, it was confirmed that the characteristics of the inter-laboratory differences were different between the hydrophilic and hydrophobic metabolites. The hydrophilic metabolites exhibited fewer inter-laboratory differences than the hydrophobic metabolites. For the hydrophilic metabolites, no association with their compound subclasses and inter-laboratory differences was observed. In the hydrophobic metabolites, the differences in the separation and MS machines affected the inter-laboratory differences. However, there was no association between their compound subclass and inter-laboratory differences. Liebisch et al. suggested the necessity of normalized protocols in lipidomics [11], and it seems to be a recent common understanding that it is important to establish global quality control samples and gold-standard approaches. However, standardization among laboratories is difficult because of the differences in the separation and MS machines, as shown in this study. In the study by Siskos et al., in the quantitative analysis of hydrophilic and hydrophobic metabolites, the standardization of the protocol led to a reduction in the differences between the laboratories [9]. Lin et al. reported that standardization after post-acquisition strategies was not effective for relative quantification precision in untargeted GC/MS metabolomics [33]. This may mean that protocol standardization is effective for the quantitative analysis of metabolomics. Background noise may easily affect the relative quantitative values. Further, the differences in ionization, depending on the separation and MS machines, may be directly linked to the results of the relative quantitation values. In particular, this study suggests that separation methods, such as GC, LC, CE, IC, and SFC, may strongly contribute to the inter-laboratory differences. It may be a difficult task to conduct the integration and subsequent evaluation of data for the metabolites analyzed as relative quantitative values by a variety of analytical methods; however, understanding the inter-laboratory differences in metabolomics results in the accurate evaluation of metabolomic data.

## 3. Materials and Methods

### 3.1. Materials

Human pooled plasma in ethylenediaminetetraacetic acid (EDTA)–2Na and mouse pooled plasma in EDTA-2Na were purchased from Kohjin Bio Co., Ltd., (Saitama, Japan) and Rockland Immunochemicals, Inc., (Limerick, PA, USA), respectively. Methanol and chloroform were purchased from Wako Pure Chemical Industries, Ltd. (Osaka, Japan). Milli-Q water (Millipore, Billerica, MA, USA) was used for all experiments.

### 3.2. Sample Preparation

Human and mouse plasma were mixed in the following proportions: Human/Mouse = 2500 µL/0 µL (S-1), 2,000 µL/2,000 µL (S-2), and 0 µL/2,500 µL (S-3). Each plasma mixture (50 µL) was mixed with 450 µL of methanol. After mixing, the mixture was kept on ice for 5 min. The solution was centrifuged at 16,000× *g* for 5 min at 4 °C, and 400 µL of the supernatant was transferred into a new tube. Afterward, 400 µL of chloroform and 200 µL of Milli-Q water were added to the supernatant and thoroughly mixed. The mixture was centrifuged at 16,000× *g* for 5 min at 4 °C; thereafter, 400 µL of the upper layer and 300 µL of the lower layer were collected into a new tube for the measurement of hydrophilic and hydrophobic metabolites, respectively. All samples were dried using a centrifuge concentrator before distribution to the six participating institutions.

### 3.3. Analytical Procedure

For the measurement of hydrophilic metabolites, GC/MS analysis was performed after oximation, and the dried samples were derivatized. Other analyses were carried out after dissolving the dried samples in water. For the measurement of hydrophobic metabolites, each analysis was performed after dissolving the dried samples in methanol or methanol:chloroform (1:1). Each measurement was carried out according to their methods for metabolomics, including the 11 analytical methods for the hydrophilic metabolites [18,21,22,23,24,25,26,27,28] and the 7 analytical methods for the hydrophobic metabolites [17,18,24,29,30,31]. The analytical methods include chromatographic separation, MS detection, and data processing; the details are presented in Appendix A.

### 3.4. Data Analysis

Data processing, including peak picking and metabolite identification, was conducted according to the specific methods of each laboratory. In the evaluations in the first step, the relative quantitative data (S-1/S-3) for each metabolite were obtained from triplicate analyses. The data processing details and normalization data for each analytical method are presented in Appendix A. The integrated data were subsequently analyzed based on a two-sided Student’s *t*-test or one-way ANOVA to evaluate if similar results were produced by multi-laboratory distinct analytical methods. The statistical significance of each metabolite between the S-1 and S-3 samples was determined using a two-sided Student’s *t*-test (*p* < 0.05). The number of metabolites with S-1/S-3 levels in the same directions among multiple analytical methods was examined using a two-sided Student’s *t*-test and a relative quantitative value of 1. Further, the number of metabolites with statistically similar S-1/S-3 levels was examined among multiple methods using one-way ANOVA (*p* > 0.05). Subsequently, one outlier in the data showing a *p*-value of < 0.05 by one-way ANOVA was found, and then the number of metabolites that changed to a *p*-value of > 0.05 by one-way ANOVA in the case that outlier was ignored was counted. All statistical analyses were performed using in-house scripts written in Python 3 using the Numpy and Scipy modules. In the second step, the average values (*n* = 3) of S-1, S-2, and S-3 ([S-1], [S-2], and [S-3]) were calculated, and the values of [S-2]/([S-1] + [S-3] × 100 were then determined. Subsequently, the same evaluations as the first step were performed using only the metabolites with values within 50 ± 7.5%, which corresponded to the 95% confidence interval when *n* = 3 and RSD% = 10%. This means that the second evaluation was performed using metabolites with guaranteed quantification and a 95% confidence interval.

## 4. Conclusions

In MS-based metabolomics, the differences in the analytical results from laboratories/machines are issues to be considered; moreover, these issues are seen in other omics technologies. The correction of inter-laboratory differences using sequential window acquisition of all theoretical mass spectra (SWATH–MS) is being attempted in proteomics [34]. Further, it seems that proteomics is proceeding in the direction of analyzing proteomic data acquired using different mass spectrometers with the same software for the data integrated evaluation. In this study, we evaluated if the differences in the analytical methods, except sample pretreatment including metabolite extraction, are involved in the inter-laboratory differences. This investigation shows that some differences between the laboratories in the current study are due to the differences in the machines only. When MS-based metabolomics systems are established at each laboratory, validation, including sample pretreatment, is carried out, and the characteristics of each machine placed in each laboratory are considered. In other words, the analytical environments and conditions at each laboratory are prioritized in the establishment of protocols. Therefore, it is difficult to resolve the inter-laboratory differences because all the laboratories cannot provide the same analytical environments. It is important to understand the problems in metabolomics studies involving multiple laboratories, and the results of this study help to achieve this understanding.

## Figures and Tables

**Figure 1 metabolites-12-00135-f001:**
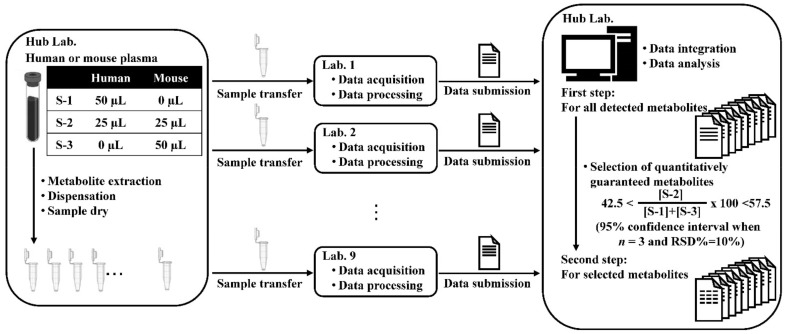
Analytical procedure in the current study. Nine laboratories (Lab. 1 to Lab. 9) participated in this study. In the hub laboratory, identically extracted samples were prepared from two kinds of plasma (human and mouse plasma), and their mixed plasma was distributed to the nine participating laboratories throughout Japan. In each laboratory, metabolomics was carried out without conducting any work other than the pretreatment required for each analysis, and 18 analytical methods were employed to extract each sample. After data submission to the hub laboratory, the data integration and data analysis were performed.

**Figure 2 metabolites-12-00135-f002:**
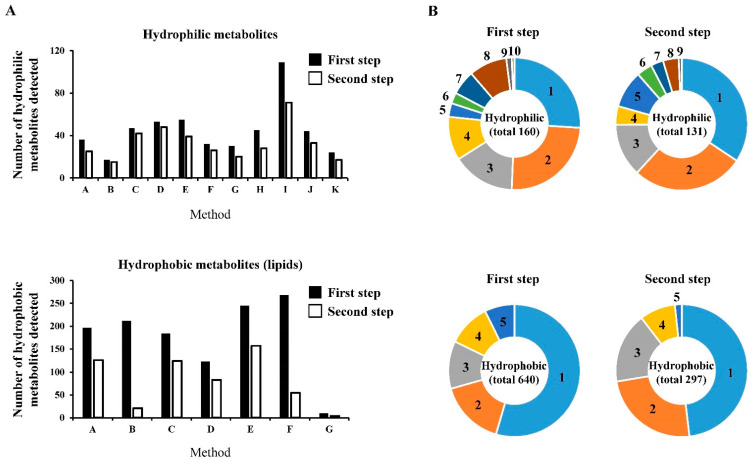
Number of metabolites detected by each analytical method and the percentages of the metabolites measured in common by multiple methods: (**A**) the numbers of metabolites detected by each analytical method for the hydrophilic or hydrophobic metabolites are shown; (**B**) the percentages of metabolites commonly measured by multiple methods are shown using pie charts. In the first step, 160 hydrophilic metabolites and 640 hydrophobic metabolites were identified in both human and mouse plasma. In the second step, 113 hydrophilic metabolites and 297 hydrophobic metabolites were identified in both human and mouse plasma.

**Figure 3 metabolites-12-00135-f003:**
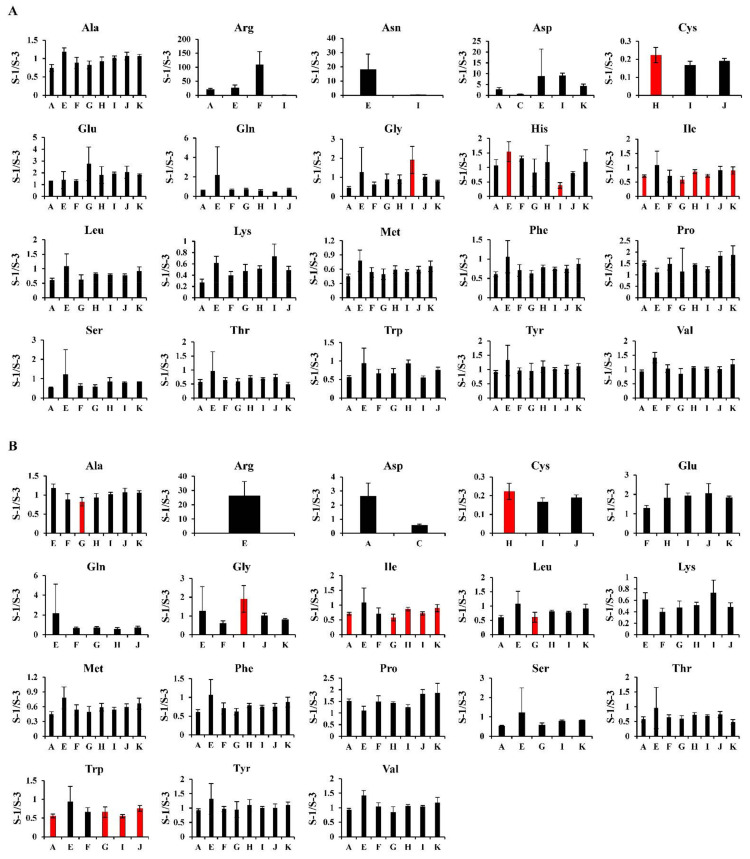
Inter-laboratory comparisons of the relative quantification for essential and non-essential amino acids. The relative quantification values (S-1/S-3) from the first (**A**) and second (**B**) steps for essential and non-essential amino acids are presented as the mean ± SD obtained from triplicate experiments. The red bars in each graph show the metabolites that were judged as the outliers based on one-way ANOVA (α = 0.05).

**Figure 4 metabolites-12-00135-f004:**
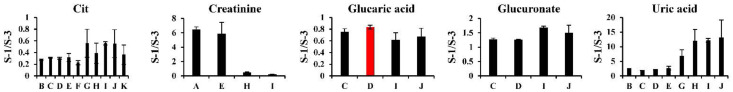
Examples of hydrophilic metabolites with remarkable differences between LC/MS, CE/MS, IC/MS, and GC/MS. As the examples of hydrophilic metabolites with remarkable differences among multiple methods (LC/MS, CE-MS, IC/MS, and GC/MS), the results of relative quantification (S-1/S-3) from the first step data treatment for citric acid (Cit), creatinine, glucaric acid, glucuronate, and uric acid are shown. The values are presented as the mean ± SD obtained from triplicate experiments. Red bars indicate outliers based on one-way ANOVA (α = 0.05).

**Figure 5 metabolites-12-00135-f005:**
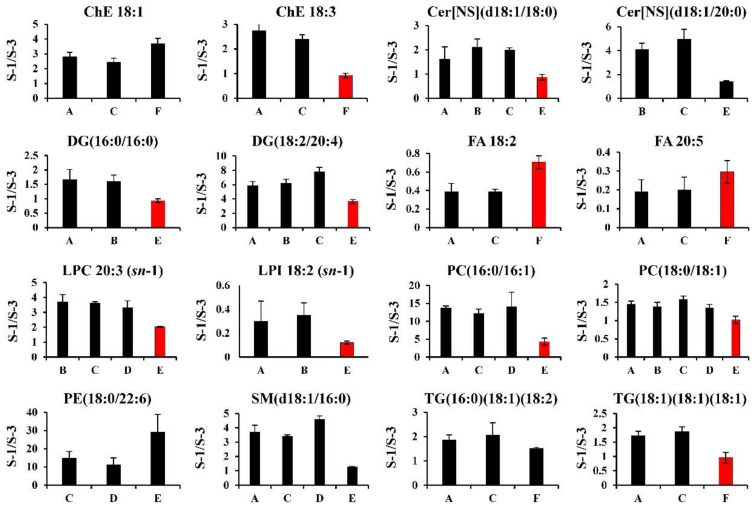
Examples of hydrophobic metabolites with remarkable differences between LC/MS and SFC/MS. As examples of hydrophobic metabolites with the remarkable differences between LC/MS and SFC/MS, the results of the relative quantification (S-1/S-3) from the first step data treatment for ChE 18:1, ChE 18:3, Cer[NS](d18:1/18:0), Cer[NS](d18:1/20:0), DG(16:0/16:0), DG(18:2/20:4), FA 18:2, FA 20:5, LP C 20:3(sn-1), LPI 18:2(sn-1), PC(16:0/16:1), PC(18:0/18:1), PE(18:0/22:6), SM(d18:1/16:0), TG(16:0) (18:1) (18:2), and TG(18:1) (18:1) (18:1) are shown. The values are presented as the mean ± SD obtained from triplicate experiments. Red bars indicate outliers based on one-way ANOVA (α = 0.05).

**Table 1 metabolites-12-00135-t001:** Analytical methods for the hydrophilic metabolites.

Method ID	Lab ID	Analytical Method & Mode	Ref.
A	1	CE–TOFMS (cation mode, scan)	[16]
B	1	CE–TOFMS (anion mode, scan)	[17]
C	1	Capillary–IC/QExactive (scan)	[18]
D	2	IC/QExactive (scan)	[19]
E	2	PFPP–LC/QExactive (scan)	[19]
F	3	C18–LC/TQMS (MRM)	[20]
G	2	Derivatization and GC/QMS (scan)	[21]
H	4	Derivatization and GC/QMS (scan)	[22]
I	3	Derivatization and GC/TQMS (MRM)	[23]
J	5	Derivatization and GC/TQMS (MRM)	[23]
K	6	Derivatization and GC/QMS (SIM)	[24]

PFPP, pentafluorophenylpropyl.

**Table 2 metabolites-12-00135-t002:** Analytical methods for the hydrophobic metabolites.

Method ID	Lab ID	Analytical Method	Ref.
A	7	C18–LC/QTOFMS (positive/negative, scan)	[25]
B	8	C18–LC/Q Exactive plus (positive/negative, scan)	–
C	9	C18–LC/Orbitrap Fusion (positive/negative, scan)	[26]
D	4	C8–LC/TQMS (positive/negative, MRM)	[27]
E	2	DEA–SFC/TQMS (positive/negative, MRM)	[28]
F	2	C18–SFC/TQMS (positive/negative, MRM)	[19]
G	3	FI/TQMS (positive, MRM)	[20]

DEA, diethylamine; and FI, flow injection.

**Table 3 metabolites-12-00135-t003:** Summary of datasets.

	Hydrophilic Metabolites	Hydrophobic Metabolites	Hydrophilic + Hydrophobic Metabolites
1. Number of identified metabolites from human plasma and/or mouse plasma samples using at least one analytical method	160	660	820
2. Number of identified metabolites from both samples using ‘two or more’ methods	111	291	402
3. Number of metabolites that were statistically significant between the human plasma and mouse plasma samples using multiple methods based on a two-sided Student’s *t*-test (α = 0.05)	88	256	344
4. Number of metabolites with similar human plasma/mouse plasma levels among the methods, based on a two-sided Student’s *t*-test (α = 0.05) and a relative quantitative value of 1	82 (93.2%)	243 (94.9%)	325 (94.5%)
5. Number of metabolites with statistically similar human plasma/mouse plasma levels among the multiple methods, using a one-way analysis of variance (ANOVA) (α = 0.05)	40 (36.0%)	62 (21.3%)	102 (25.4%)
6. Number of metabolites with statistically similar human plasma/mouse plasma levels among the multiple methods, ignoring one outlier method using a one-way ANOVA (α = 0.05)	56 (50.5%)	135 (46.4%)	191 (47.5%)

The percentages (%) in the parentheses in items 5 and 6 indicate the ratio of the numbers in item 5 or 6 to item 2. The percentages (%) in parentheses in item 4 indicate the ratio of the numbers in item 4 to item 3.

**Table 4 metabolites-12-00135-t004:** Summary of the datasets after the selection of the quantitatively guaranteed metabolites.

	Hydrophilic Metabolites	Hydrophobic Metabolites	Hydrophilic + Hydrophobic Metabolites
1. Number of identified metabolites from human plasma and/or mouse plasma samples by at least one analytical method	131	297	428
2. Number of identified metabolites from both samples by ‘two or more’ multiple methods	86	154	240
3. Number of metabolites that were statistically significant between the human plasma and mouse plasma samples from multiple methods based on a two-sided Student’s *t*-test (α = 0.05)	66	123	189
4. Number of metabolites that showed similar human plasma/mouse plasma levels among the methods based on a two-sided Student’s *t*-test (α = 0.05) and a relative quantitative value of 1	60 (90.9%)	117 (95.1%)	177 (93.7%)
5. Number of metabolites with statistically similar human plasma/mouse plasma levels among multiple methods using a one-way ANOVA (α = 0.05)	30 (34.9%)	49 (31.8%)	79 (32.9%)
6. Number of metabolites with statistically similar human plasma/mouse plasma levels among multiple methods, ignoring one outlier using a one-way ANOVA (α = 0.05)	48 (55.8%)	87 (56.5%)	135 (56.3%)

The percentages (%) in the parentheses in items 5 and 6 indicate the ratio of the numbers in item 5 or 6 to column 2. The percentages (%) in the parentheses in item 4 indicate the ratio of the numbers in item 4 to item 3.

## Data Availability

The data presented in this study are available within the article and Appendix A. All tables and figures are original and have not been taken from any publication.

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
