# Peer review of "Comparative Evaluation of Plasma Metabolomic Data from Multiple Laboratories"

_metabolites, 2022, doi:10.3390/metabo12020135_

Round 1
Reviewer 1 Report
This is an excellent paper in my opinion. It hi-lites the issues which could befall large metabolite studies needing multiple machines/ laboratories. As this paper states there can be significant differences in the metabolites, including quantitive measurements. It is a warning. As the authors observe, standardisation can lead to a reduction between differences in laboratories/machines, although its may not be sufficient.
Although the issues have been well described, perhaps there could be a bit more written, added, on solutions, to help workers in the field, rather than describing the issues/problems. hence I suggest minor revision.
Its an extremely well written paper, the English is excellent.
Reviewer 2 Report
Manuscript title: Comparative evaluation of plasma metabolomic data from multiple laboratories.
General comment: In this paper the authors investigated the effects of several factors, among which measurement, data analysis and sample preparation on the inter-laboratory differences in metabolomic data with the aim of facilitating the understanding of the problems in metabolomics studies involving multiple laboratories. Before publication on Metabolimics, I suggest minor revisions.
L104-L109: I suggest the authors to better explain the main goal of their work at the end of the Introduction section, since it’s not enough clear.
Section 2.1. The authors title this section “Experimental design”. This can be misleading, as experimental design or DoE is typically referred to experiments aim at predicting the outcome by introducing a change of the preconditions, which is represented by one or more independent variables.
Moreover, in this section they should clearly report the number of laboratories involved in the study and the differences among the methods that they used.
L153: “[S-2]/([S-1]+[S-3])×100” Please, clarify what S stands for.
Figure 1: The figure is informative since it helps the reader to understand the operating procedure of the experiments. However, the authors should at least indicate the total number of laboratories that participated in the study.
Table 3 and Table 4. “The percentages (%) in the parentheses in columns 5 and 6 indicate the ratio of the numbers in column 5 or 6 to column 2.” Only 4 columns are reported in these tables. Please, carefully check.
L320: “Many of the relative ratios (human/mouse) were lower in SFC/MS (Methods E–F) than in LC/MS (Methods A–D)..” I suggest the authors to argue this point, as in general for some classes of compounds such as alkyl polyglyceryl ethers SFC offer a better separation compared to LC, which coupled with MS provided very useful structural information, ensuring also more sensitive results in quantification. To this regard, see and cite (2021). Analysis of surfactants by mass spectrometry: Coming to grips with their diversity. MASS SPECTROMETRY REVIEWS, (2021). p. 1-32, ISSN: 0277-7037, doi: 10.1002/mas.21735.
